# Origin, Evolution, and Research Development of Donkeys

**DOI:** 10.3390/genes13111945

**Published:** 2022-10-25

**Authors:** Yonghui Wang, Xiaopeng Hua, Xiaoyuan Shi, Changfa Wang

**Affiliations:** Liaocheng Research Institute of Donkey High-Efficiency Breeding and Ecological Feeding, Liaocheng University, No.1 Hunan Road, Dongchangfu District, Liaocheng 252059, China

**Keywords:** domestic donkey, origin of domestication, bioarcheology, gene sequence

## Abstract

Lack of archaeological and whole-genome diversity data has restricted current knowledge of the evolutionary history of donkeys. With the advancement of science and technology, the discovery of archaeological evidence, the development of molecular genetics, and the improvement of whole-genome sequencing technology, the in-depth understanding of the origin and domestication of donkeys has been enhanced, however. Given the lack of systematic research, the present study carefully screened and collected multiple academic papers and books, journals, and literature on donkeys over the past 15 years. The origin and domestication of donkeys are reviewed in this paper from the aspects of basic information, cultural origin, bioarcheology, mitochondrial and chromosomal microsatellite sequences, and whole-genome sequence comparison. It also highlights and reviews genome assembly technology, by assembling the genome of an individual organism and comparing it with related sample genomes, which can be used to produce more accurate results through big data statistics, analysis, and computational correlation models. Background: The donkey industry in the world and especially in China is developing rapidly, and donkey farming is transforming gradually from the family farming model to large-scale, intensive, and integrated industrial operations, which could ensure the stability of product quality and quantity. However, theoretical research on donkey breeding and its technical development lags far behind that of other livestock, thereby limiting its industrial development. This review provides holistic information for the donkey industry and researchers, that could promote theoretical research, genomic selection (GS), and reproductive management of the donkey population.

## 1. Introduction

The donkey is an equine animal of the order Odd-toed ungulates. Among common livestock animals, it occupies an important place in the history of human transportation. It is often used as a pack animal for long-distance transportation due to its good packability, durability, and traction, thereby greatly facilitating commerce among regions. With the advancement of modern technology, mechanization of agriculture, and rapid development of transportation, the service value of donkeys has decreased. In economically developed regions (such as Italy and America), donkeys are more often bred as ornamental pets. However, in fast-growing developing countries (such as China), donkeys are more often used as a source of meat for mass breeding because of their rich intra-muscular fat and superior taste [1]. In recent years, donkey milk is being sold as a commodity for the treatment of infants, the sick and the elderly who are physically weak. Donkey milk is rich in nutrients and is whey-protein milk, which is the closest to human milk and can be used as a substitute for breast milk, and has various functions such as regulating immunity and promoting growth [2]. The scale of donkey breeding and the quality of donkey breeds are also decreasing in developed areas; however, the size of donkey inventories continues to slowly increase in developing areas [3].

According to incomplete statistics (FAOSTAT, Live Animals data: https://www.fao.org/agroecology/database/detail/en/c/1473377/ (accessed on 1 January 2019)), the total number of donkeys worldwide is approximately 50 million. The highest number of donkeys in China was about 12 million, but by 2019, donkeys faced a gradual decline in breeding stock to 2.6 million (China Statistical Report, 2019). The number of donkeys in Ethiopia has risen from 5 million to over 8 million, making it the country with the largest number of donkeys in the world (Food and Agriculture Organization of the United Nations). Sub-Saharan Africa, Northern India, and the tropical highlands in Latin America have been the main areas of increased donkey populations over the past three decades, while the Mediterranean region has experienced a decline, and the overall trend has been moderate growth [4,5].

### 1.1. Methodology

The literature was searched using Google Scholar, Web of Science, Elsevier, and Endnote search engines, with keywords including but not limited to domestic donkey, origin, domestication, bioarcheology, and gene sequence, respectively. The research carefully screened and collected multiple academic papers and books, journals, and literature on donkeys over the past 15 years. The selection criteria of references were related to the content of origin, domestication, evolution, and research into the donkey genome. Key references were finally identified through repeated readings and comparisons.

### 1.2. Ethical Statement

The Animal Welfare and Ethics Committee of the Institute of Animal Sciences, Liaocheng University (No. LC2019-1).

## 2. Origin of the Donkey and Archaeological Findings

### 2.1. Linguistic Evidence on the Origins of the Donkey

The early history of the African donkey is arduous to explore due to the lack of direct archaeological data. The wild relatives of the donkey have been hunted to extinction. Therefore, gathering detailed data to track the genetic information of their ancestors is difficult. Another strategy for filling this historical gap is the use of linguistics. Donkey and donkey terminologies have been recorded in various African and Near Eastern languages. Compiling these terms and tracing the connections could suggest some hypotheses about the domestication process and its pathways of transmission. These connections could be combined with modern ethnographic data to reconstruct the prehistory of the African donkey [6,7].

Linguistic evidence suggested that various branches of West Asian-African languages appear to have fairly distinctive wild-ass vocabularies. The main distributions are as follows: #kuur, which is widely distributed in Africa; #harre Ethiopian; #d-q-r Cushitic; #aɣyul Berber; and #aʒḍ Berber. In most cases, people were sufficiently familiar with the wild ass to name the creature in pre-domestication times. A comprehensive analysis revealed that the root #k-r is common in Central Africa, and it appears to have been transmitted to the Lake Chad region from the Cushitic region of the Horn of Africa (thus subsumed under the Nilo-Saharan languages). These findings prove that donkeys have been described in various languages of different tribes at different times, leading to the hypothesis that donkeys may have been domesticated many times on the Sahara periphery [8,9].

### 2.2. Archaeological Discovery of the Domestic Donkey

In the early years of human society, small city states proliferated, and the political, economic, social, and ritual natures of cities experienced an extensive change. Production specialization, large-scale long-distance transportation, and the extensive increase in the scale of warfare led to an increasingly important role of donkeys in the exchange between regions. Goods, such as copper and other products, were increasingly transported. Development that transformed the scale of the economic system benefited the domestic asset owners, creating a new class of merchants, and donkeys became progressively valuable. As donkeys began to be widely used in the Near East, “donkey caravans” were formed to specialize in the transportation of goods. Donkeys spread as a totem associated with worship among merchants and herders, who occupied specialized positions in the growing complexity of the social structure of urban societies, to secure their social status. An intact fossilized donkey bone was found at a site in Israel. The results of the burial excavation and analysis were used to integrate zooarchaeological, architectural, stratigraphic, and typological analyses of this bioarcheological deposit and identified donkey burial as a ritual deposit, clarifying the importance of this taxon to the religious and economic spheres of the Near East Early Bronze Age (EBA). If the results are confirmed, these findings will open a new gateway to understanding donkey burials scattered throughout the region [10,11].

### 2.3. Archeopathological Study of the Domestic Donkey 

The domestication process in donkeys may be slower and not as linear as previously supposed. Species–specific indicators of the early stages of domestication need to be identified to characterize the length and course of this complex process. Unfortunately, the earliest stages of donkey domestication are when they most resemble their wild ancestors, making them difficult to distinguish. The recently discovered skeleton and concurrent studies of modern African wild ass and donkey metacarpals were intended to elaborate the analysis of the original skeleton and develop markers for the domestication process of the donkey. Skeletal fragments show a tendency for Egyptian donkeys to decrease in size over time; thus, skeletologically, large donkeys are usually considered wild and small donkeys are usually considered domestic [12]. Morphological evidence of load-carrying is displayed on contemporaneous burial remains from the same region, providing powerful support for domestication [13]. The well-preserved integral skeleton provides a morphological context for each bone that isolated skeletal fragments from previous archaeological findings lack, and a particular opportunity to apply metacarpal indicators and new paleopathological methods to distinguish hunted wild asses from domesticated ones [9,14].

In 2011, a complete donkey skeleton was found at the Xinjie site in Lantian, Shaanxi Province, China (Late Yangshao and Early Longshan cultures, ca. 4350–3950 years ago). It was buried in a foundation pit at the house base, and archaeologists extracted the complete skeleton. Archaeologists further analyzed the bones and confirmed that the donkey had been domesticated, making it the earliest domesticated donkey found in China. This finding illustrated that donkeys have been raised in China for about 4000 years [15]. Comparative analysis of the surface traces of animal bones showed that these traces were mainly chopped, cut, having carnivorous gnawing marks, and no bone abrasion by carrying goods was found on the sides of the lumbar vertebrae. These bones were mainly from large and medium-sized animals hunted by ancient humans, such as the Mongolian wild ass (*Equus hemionus hemionus*), water buffalo (*Bubalus bubalis*) and Przewalski’s gazelle (*Procapra przewalskii*). They prioritized the exploitation and use of nutrients from these large- and medium-sized herbivores (especially Mongolian wild ass) over domestication and load-carrying [16]. Mongolian wild asses were not domesticated by ancient humans during this period [15]. Although the Chinese domestic donkeys and Mongolian wild asses have a high similarity in some appearance characteristics, the chromosome numbers of the Chinese domestic donkey and Mongolian wild ass are different. Molecular evidence demonstrates that Asiatic wild ass (*Equus hemionus*) is not the ancestor of the Chinese domestic donkey and they are related as two different species from the same genus [17,18]. These findings indicated that the domestic donkey did not evolve from the Mongolian wild ass, and their similar physical appearance is probably due the same ambiance and convergent evolution.

## 3. Research Findings at the Molecular Cellular Level

### 3.1. Chromosomes

Horses, donkeys, and zebras belong to the genus *Equus*, which was formed approximately 4–4.5 million years ago. Although the equine fossil record represents a textbook example of evolution, the sequence of events that led to the existence of species diversity to date remains unclear. The entire genome and the genomes of surviving equine species were sequenced, the genetic material composition of specific genealogies were deciphering, and the complex history of the formation of equine species was revealed. Surprisingly, multiple examples of hybridization have been found throughout the genus *Equus* despite extremely different chromosomal structures, in contrast to theories that promoted chromosomal incompatibilities as a driving factor in the origin of the equine species [19,20].

Six indexed DNA libraries from the Somali wild ass (SOM), Onager (ONA), Tibetan kiang (KIA), Grevy’s (GRE), mountain (HAR), and plains (BOE) zebras were prepared successfully. The genome sets of all extant zebras and donkeys of the genus are currently completely represented. In addition, museum specimens were used to characterize the genome of extinct quagga zebra (which became extinct in the early 1900s) [21]. Combined with the horse genome data from previous studies [22], this completes the genomes of all extant species of the genus. The analysis was performed through simulation experiments by scanning all genomes. The results showed that the earliest species formation gene flow occurred in North America. The ancestors of today’s donkeys and zebras dispersed between 2.1 and 3.4 million years into the American continent, eventually experiencing major population expansions and collapses that coincide with past climate change events. Evidence of gene flow involving three contemporary equine species was also found despite the chromosome numbers ranging from 16 pairs to 31 pairs. These findings challenged the notion that the accumulation of chromosomal rearrangements drove complete reproductive isolation and promoted equids as a fundamental model for understanding the interplay between chromosomal structure, gene flow, and eventual species formation [23].

### 3.2. Fluorescence In-Situ Hybridization (FISH)

Chromosome banding and new molecular cytogenetic techniques, particularly FISH, using chromosome painting probes, were used early on to understand the phylogenetic and systemic relationships of species and provide insights into the possible mechanisms of species formation [24]. Comparative detection was performed on the sequences of telomeres, meristematic granules, and kernel composition regions in domestic horses (*Equus caballus*) and domestic donkeys (*Equus asinus*) by using primed in-situ DNA synthesis (PRINS) and FISH. The findings did not reveal any additional sites in horses and donkeys but confirmed the differences in signal intensity and frequency between the individual chromosome pairs in the two species. As in horses, no interstitial telomeric sites (ITSs) were detected in the donkey genome through PRINS analysis, possibly due to the multiple chromosomal rearrangements or the gradual loss of repetitive sequences that occurred during evolution after divergence from a common ancestor. On the contrary, another hypothesis indicated that these sequences are present in a very low copy number, hence not detected [25,26].

The presence of constitutive alkaline disruption sites (ALSs) in donkey (*E. asinus*) and stallion (*E. caballus*) spermatozoa was investigated using DNA breakage detection–FISH and comet assay. ALSs in the sperm of donkey was 1.3 times greater than in stallion and the length of the comet tail obtained in donkey sperm was 1.6 times longer than that observed in horse (*p* < 0.05). The difference is significant between these two species. The results suggested that ALS represents a species-specific issue in mammalian species related to chromatin organization in sperm and somatic cells, and it may diverge even at short phylogenetic distances [27].

Within the karyotype of the domestic donkey, heterochromatic bands of non-centric chromosomes have been described in sub-centric and telomeric positions. By using FISH, changes in the intensity and distribution of fluorescent signals were observed after in-situ hybridization with two DNA probes containing fragments of the two major equine satellite DNA families. Chromosome 1 has heterochromatic bands in the proximal regions of the long and short arms, and the number and distribution of large clusters of satellite DNA could define at least nine polymorphic variants constituting sexual heterochromatin that could not be detected by the C-banding method alone [28]. 

## 4. Research on Genetic Material

### 4.1. Ancient DNA

A low-coverage draft genome sequence was acquired from a horse bone recovered from a permafrost site in the Yukon Territory, Canada, dated to approximately 560,000 to 780,000 years ago [29]. Comparative genomics revealed that the *Equus* lineage that gave rise to all contemporary horses, zebras, and donkeys originated approximately 4–4.5 million years ago, which is much earlier than previously suspected. These data supported the contention that Przewalski horses, an endangered subspecies native to the Mongolian steppe, represent the last surviving population of wild horses.

The available genetic data from donkeys revealed two distinct mitochondrial DNA (mtDNA) haplotypes, indicating two distinct domestication events that occurred in Northeastern Africa approximately 5000–7000 years ago. The absence of a clear phylogeographic structure in domestic donkey haplotypes and the paucity of information on the genetic composition of African wild ass ancestors resulted in difficulty determining the feral ancestry and geographic origin of domestic mitochondrial branches. The analysis of ancient archaeological and historical museum samples provided genetic information on the historical Nubian wild ass (*Equus africanus africanus*), Somali wild ass (*Equus africanus somaliensis*), and ancient donkeys. The results indicated that the donkeys of clade 1 have a long history in the Sahara, and their ancestors were Nubian wild asses that were crossed with domestic donkeys over a long period of time by introducing several maternal haplotypes from the wild asses; the gene flow of this clade is continuous. A relative of the Somali wild ass is the ancestor of clade 2 (probably extinct). The Somali wild ass, on the other hand, belongs to clade 3, which is highly separated from clades 1 and 2, indicating that the Somali wild ass is not the ancestor of these two clades. The low variation and large sample size of the Somali wild ass made it unlikely that other lineages could be identified [30]. The very ancient merging period of the evolutionary branch reflects a long period before donkey domestication, suggesting that the extensive genetic structure, fragmentation, and geographic isolation of mitochondrial variation in wild asses may have preceded domestication. The results illustrated the complexity of animal domestication and valuably contributed to the debate on the variation, phylogeny, and management of the extant but critically endangered African wild ass. Most of the available mitochondrial diversity samples have been obtained from captive-bred Somali wild-ass populations, but further research of the DNA and chromosomes of extant populations requires additional specimens for DNA analysis.

In 2022, Todd et al. constructed a comprehensive genomic panel of 207 modern and 31 ancient donkeys and 15 wild equids in order to elucidate the domestication history of donkeys, uncovering a robust phylogeographic structure of modern donkeys [31]. The findings support a single domestication of donkeys in Africa around 7000 years ago, followed by further expansion across that continent and Eurasia and eventual return to Africa. The latest findings are not consistent with previous studies, with differences in the number of domestications of donkeys. In addition, researchers have discovered a previously unknown genetic lineage in the Levant (2200 years ago), a finding that adds to the ancestry of the Asian donkey.

Evolutionary processes, including selection, could be indirectly inferred from the patterns of genomic variation in contemporary populations or species. Sequencing ancient DNA from samples with time intervals could provide insights into the past selection processes, as time-series data could directly quantify population parameters collected before, during, and after selection-driven genetic changes [32]. Incorporating temporal sampling and generation of ancient genomic datasets in the context of evolutionary biology and using some emerging techniques that have not been widely used by evolutionary biologists more accurately restore the true population data of the species throughout its history. However, these same data have limitations, and they may be influenced by post-mortem damage, fragmentation, low coverage, and typically low sample sizes.

### 4.2. Microsatellite Markers

Within the framework of varietal protection, genetic characteristics are important for the integrity of the variety, and they are a prerequisite for the treatment of genetic resources. In the last decades, the use of molecular markers has played an important role in the analysis of genetic diversity and genetics. Among the different types of molecular markers, microsatellite markers are widely used due to their ease of PCR amplification and ability to analyze large amounts of genetic variation (allelic variation at each locus). Microsatellite markers are simple sequence repeats, usually consisting of 1–6 nucleotide repeats, which are abundant and distributed throughout the genome. Microsatellite markers are highly polymorphic, species-specific, and co-dominant compared with other molecular markers, and thus have become increasingly important genetic markers for genetic diversity, population genetics, and disease diagnosis. These advantages validate the continued use of microsatellite markers in different studies to quantify genetic variation within and between species for the conservation management of animal populations [33,34].

Studies on the genetic characteristics of donkey breeds is scarce, and they are mainly concentrated on Mediterranean and Asian breeds. In 2006, 24 pairs of microsatellites from the horse genome were used to amplify the genomes of eight local donkey breeds in China. The results indicated that the microsatellites of similar species are conserved and could be used to analyze the genetic diversity of donkeys. Researchers also needed to have a deeper understanding of the genetic information and genetic relationships within and among breeds to obtain more accurate and generalized conclusions, which could be more effectively applied to the conservation and further utilization of donkey breeds [35,36,37]. In terms of average allele number, the genetic variability observed in three Sicilian donkey breeds should be lower than in five Spanish and three Croatian breeds but higher than in the Italian Amiata donkey [38]. However, the expected heterozygosity was lower than that of the European breeds and the eight Chinese donkey breeds mentioned above. Considering that the breed Pantesco is undergoing genetic recovery, the actual numbers are sparse, thus making it an endangered breed with low genetic variability. Molecular characterization of Sicilian varieties revealed a high degree of internal structure, evidence that could be largely attributed to the Pantesco structure, which is clearly distinct from Ragusano and Grigio Siciliano; meanwhile, the significant divergence of the Pantesco structure seems to be the hallmark of the proper genetic management program undertaken thus far [39,40].

In the past decade, researchers in various countries around the world, such as the United States [41], India [42], South Korea [43], Ethiopia [44], Turkey [45], and Serbia [46] have also been using microsatellite markers to experimentally test local donkey breeds. However, with the exception of China and Italy, these studies have not formed a coherent research system, partly related to the endangered population of local donkey breeds, the weak emphasis on donkey science, and the lack of research efforts to protect local donkey breeds. The Mongolian wild ass [47] and the Kiang (*Equus kiang*) in China, and the Pantesco, Ragusano , and Grigio Siciliano in Italy [40], and Banat donkey in Serbia [46] are protected as endangered species, and their population size has been increased to some extent.

## 5. Sequencing and Assembly of the Genome

### 5.1. Mitochondrial DNA (MtDNA)

Mitochondrial DNA (MtDNA) is maternally inherited so that any maternally related individual could have the same mtDNA sequence. In contrast to the more traditional nuclear DNA markers typically used, mtDNA provides a valuable locus for forensic DNA typing in some cases. With technological advances, researchers in molecular laboratories have developed new assays. Long-term dead or missing individuals or any living maternal relatives could provide a reference sample that is extremely useful in determining the identity of an individual. Furthermore, a large number of nucleotide polymorphisms or sequence variants in two highly variable portions of the non-coding control region could be effective in distinguishing between individuals and/or biological samples. Muscle, bone, hair, skin, blood, and other body fluids may provide sufficient material to type the mtDNA locus despite degradation due to environmental stress or time. MtDNA is inherited only from the mother: thus, any maternally related individual could provide a reference sample in cases where direct comparison with biological samples is not possible. However, the maternal inheritance pattern of mtDNA may also be considered problematic. As all individuals in a maternal lineage have the same mtDNA sequence, mtDNA could not be considered a unique identifier. Indeed, apparently unrelated individuals may have shared an unknown maternal relative at some distant point in the past. 

The donkey is the only individually domesticated hoofed animal in Africa. Its origin was assessed by sampling donkeys from 52 countries in the Eastern hemisphere and sequencing 479 base pairs (bp) of the mtDNA control region [48]. Two highly divergent phylogenetic groups were systematically analyzed and identified: Asian wild half-asses (*E. hemones* and *E. kiang*) and two extant wild African ass subspecies, the Somali wild ass (*E. africanus somaliensis*) and the Nubian wild ass (*E. africanus africanus*). The findings ruled out the possibility of Asian wild half-asses as an ancestor of the Chinese donkey. The African wild ass is anticipated to be an ancestor of the Chinese donkey. Research showed that the practice of animal domestication first emerged in the Near East but re-emerged in Northeastern Africa, a region that may have been particularly instrumental in the expansion of population and trade in the Old World. The domestication of donkeys may have originated as a response to Saharan desertification (5000–7000 years ago) by pastoralists and other social elements in Northeastern Africa, and it supplies clues for archaeological studies looking for evidence of the original domestication of the donkey.

In 2017, Stanisic et al. assessed the current genetic status of the three largest populations of E. asinus in the central Balkans (Serbia) by analyzing the variability of the nuclear microsatellite and mitochondrial (mtDNA) control regions of 77 and 49 individuals, respectively [49]. A comparative analysis of mtDNA datasets and mtDNA sequences of 209 published ancient and modern individuals from 19 European and African populations provided new insights into the origin and history of Balkan donkeys. The Balkan donkeys (*Equus asinus* L.) in Serbia are diverse, with populations that are highly genetically diverse at the nuclear and mtDNA levels, but with severely declining populations. The two groups of individuals were found to have similar phenotypic characteristics, different nuclear backgrounds and different proportions of mtDNA haplotypes belonging to maternal Clades 1 and 2. Clade 2 may have appeared in Greece before Clade 1, while the expansion and diversification of Clade 1 in the Balkans preceded that of Clade 2.

MtDNA D-loop and cytochrome b gene fragments were amplified and sequenced from 21 suspected donkey remains from four archaeological sites in China to explore the matrilineal origin and transmission routes of the Chinese donkey [18]. Phylogenetic analysis revealed that ancient Chinese donkeys had a high mitochondrial DNA diversity and two distinct mitochondrial maternal lineages, Somali and Nubian. The results implied that the maternal origins of Chinese domestic donkeys may be related to African wild asses (which include Nubian and Somali wild asses), and along with historical records, showed that these origins were introduced to Western and Northern China before the appearance of the Han Dynasty (2202 years ago). During the Tang Dynasty (618–907), when the Silk Road reached its golden age, domestic donkey populations in China increased, primarily to meet the demands of expanding trade. These donkeys were likely used as commodities or for the transportation of goods along the Silk Road. The research provides valuable ancient animal DNA evidence for early trade between African and Asian populations for the first time. DNA analysis of ancient Chinese donkeys revealed the dynamics of matrilineal origins, domestication, and transmission pathways.

A total of 367 mtDNA D-loop sequences of Chinese donkeys were analyzed by scholars, and 96 haplotypes and 57 polymorphic loci were found to have a rich genetic diversity by analyzing the experimental results [50]. Considering that the mtDNA genetic diversity is distributed among all donkey breeds and sizes, no obvious relationship exists between the matrilineal inheritance of donkeys and their geographical distribution or body size. Moreover, the sequences of Chinese and Asian wild asses are not clustered together, thus ruling out the possibility that the Asian wild asses are the maternal ancestor of the Chinese donkey. As for the similarity in fur color and morphology between some Chinese domestic donkeys and Asian wild asses, it may be caused by the convergent evolution of these two species living in a similar ecological environment. As the fur color of donkeys is determined by nuclear genes, whether homology exists between the fur color genes of domestic donkeys and wild asses remains to be investigated.

By using mtDNA Cyt b gene sequences and Y chromosome microsatellite methods, the genetic diversity and origin of 273 male donkeys from 13 domestic donkey breeds in China were studied [51]. The results showed that the Chinese donkey has a rich genetic diversity in the Cyt b gene. No polymorphism was found at any of the five Y-chromosome-specific microsatellite loci, showing that the Y-chromosome genetic diversity of the Chinese donkey is extremely low and only one paternal origin exists. Two of the Y-chromosome microsatellites could be used as microsatellite markers to distinguish the Chinese domestic donkey from the European domestic donkey, indicating that these donkeys may have different paternal origins. The low mutation rate of the Y chromosome in the Chinese donkey contrasts with the high mutation rate of the mitochondrial D-loop region and the rich genetic diversity of the autosomal microsatellites, suggesting a greater role for matrilineal inheritance in genetic diversity in the Chinese donkey. A strong gender bias may also be present in early breeding, where one male donkey breeds with multiple females. 

The genetic diversity, origin, and domestication of donkeys have been extensively studied using autosomal microsatellites and mitochondrial genomes. However, the male-specific regions of the Y chromosome of modern donkeys are largely uncharacterized. Fourteen published Y-chromosome-specific microsatellite (Y-STR) studies were performed on 395 male donkey samples from China, Egypt, Spain, and Peru by fluorescently labeling microsatellite markers [52]. The results indicated seven male-specific polymorphisms and showed 2–8 alleles with polymorphism in donkeys. A total of 21 haplotypes were identified, possibly reflecting weak sex bias in breeding programs, with a large amount of paternal inheritance contributing to the high male effective population size of native Chinese donkeys. Three haplotype groups were also identified, indicating three separate paternal lines in domestic donkeys. The five Y-STR markers in donkeys showed polymorphism. The Y-STR of donkeys is richer in genetic diversity than that of horses, and the relatively high level of Y-chromosome variability is consistent with the extensive mtDNA diversity of domestic donkeys. The abundance of polymorphisms is a common feature of donkey nuclear DNA, particularly the abundance of microsatellite polymorphisms on autosomes. These markers could be used in studies of donkey Y chromosome diversity and population genetics in African, European, South American, and Chinese donkeys.

### 5.2. Whole Genome Sequencing (WGS)

WGS is a comprehensive method for analyzing entire genomes. It could detect single nucleotide variants, insertions/deletions, copy number changes, and large structural variants. Due to the recent technological innovations, the latest genome sequencers can perform WGS more efficiently than ever. Unlike focused approaches, such as exome sequencing or targeted resequencing, which both analyze a limited portion of the genome, WGS delivers a high-resolution, base-by-base view of the genome. Firstly, it could capture large and small variants that may be missed with targeted approaches. Then, it could identify potential causative variants for further follow-up studies of gene expression and regulation mechanisms. Lastly, it could deliver large volumes of data in a short amount of time to support the assembly of novel genomes. Although WGS has many advantages, it also has several disadvantages that should be mentioned. First, the role of most genes in the genome is still unknown or incompletely understood, indicating that some “information” found in the genome sequence is unusable at present. Second, an individual’s genome may contain information that the individual does not want to be known. Third, the volume of information contained in a genome sequence is vast, and analyzing these data is difficult. The scalable, flexible nature of next-generation sequencing technology makes it equally useful for sequencing any species, such as livestock, plants, or disease-related microbes. The rapidly dropping sequencing costs and ability to produce large volumes of data with today’s sequencers make WGS a powerful tool for genomics research.

Compared with other mammals, the species of genus *Equus* have a more pronounced karyotypic diversity, and they are high-quality models for exploring karyotypic instability. The high frequency of mitogenic repositioning events in this genus is a puzzling phenomenon, and analysis of whole-genome sequences is a sophisticated and powerful method to study chromosome evolution. The findings of whole-genome assembly from donkeys and Asian wild asses could reflect the donkey’s unique characteristics, that is, more efficient energy metabolism and better immunity than horses [53]. Researchers detected abundant satellite sequences in some inactive meristematic regions but not in neo-meristematic regions. On the contrary, ribosomal RNA was frequently present in neo-meristematic regions rather than obsolete meristematic regions. The donkey and Asian wild-ass genomes complement the reference genomes of the genus *Equus*, and comparative analyses based on these genome sequences provided important insights into the demographic history and adaptive evolution of the donkey. Furthermore, these results enhanced the understanding of chromosomal rearrangements and characteristic sequence dynamics associated with filament repositioning; the data could contribute to future studies of genomics and mammalian chromosome evolution in the genus *Equus*.

### 5.3. Genome Assembly—Dovetail Chicago Technology

Long-range and highly accurate de-novo assembly from short-read data is one of the most pressing challenges in genomics. Read pairs generated by proximity ligation of DNA in living tissue chromatin could solve this problem, thus extensively improving the scaffolding continuity of the assembly. In 2016, Putnam et al. reported an in-vitro method for generating long-distance ligation data that improves the scaffolding of de-novo assembled genomes [54]. Their methodology, called Chicago, requires only a small amount of high molecular weight DNA as starting material and uses recombinant chromatin as a substrate to generate a proximity ligation library. By using HiRise software, genomic scaffolds could be generated. The authors demonstrated the utility of their methodology for human genome assembly and scaffolding. Moreover, the Chicago library could be used to improve existing assemblies, as illustrated by reassembling and building genomic scaffolds of American alligators and for haplotype staging. The main weakness linked to HiRise is the introduction of assembly errors (mostly short indels) that must be corrected with accurate short reads. The sequencing errors could hinder the check accuracy gene annotations (BUSCO) and protein prediction.

Donkeys and horses share a common ancestor of the genus that lived 4.0–4.5 million years ago. While high-quality genome assembly for horses at the chromosomal level is available, the current genome assembly for donkeys is limited to moderate scaffold size. The novel high-quality donkey genome assembly obtained using HiRise assembly technology, which provides scaffolds of higher quality and length (sub-chromosomal size) than the existing donkey genome assemblies, further expanded the studies of selective breeding, equine evolution (including species formation and domestication), and conservation. The new assembly could identify runs of homozygosity (ROHs) caused by low effective population size and ancestral affinities, thus exploring their impact on the existing distance patterns between donkeys and horses [55]. This new assembly was used to obtain more precise measures of heterozygosity (genome wide and local) than horses and detect donkey purity that may be associated with positive selection. It was able to identify fine chromosomal rearrangements between horses and donkeys which may have played an active role in their differentiation and ultimately, speciation.

Recently, researchers assembled a new draft genome of the Kiang and performed a large-scale resequencing of the Kiang and domestic donkey genomes [56]. The findings show that Kiang and Tibetan donkeys utilize different genes (*EPAS1* and *EGLN1*, respectively) to adapt to low oxygen conditions associated with living at high altitude. Both genes, *EPAS1* and *EGLN1*, are the two most important genes for high altitude adaptation in Tibetan and other highland animals, indicating that the number of potential biological pathways involved in high altitude adaptation in mammals may be limited. The results of the comparative analysis revealed that the Tibetan donkeys did not acquire the ability to withstand high altitude through adaptive evolution with the Kiang, and gene introgression between the two is rare. On one hand, it may be because Kiang and Tibetan donkeys’ hybrids cannot reproduce offspring. On the other hand, it may also be because donkeys live on the Tibetan plateau for a short period of time and have limited encounters, making it difficult to generate gene flow. Given their biological similarities, EGLN1 and EPAS1 could provide markers for donkeys bred at other high altitudes in the world.

### 5.4. Genome Assembly—Hi-C Technology

Hi-C sequencing is based on chromosome conformation capture technology (3C), which applies high-throughput sequencing to capture spatially contiguous fragments across the entire genome and reveals three chromatin conformations in the nucleus. In descending order, the three-dimensional (3D) hierarchical structural units of the mammalian genome are chromosome territory, chromatin compartment A/B, topological associated domain, and chromatin loop. The advancement of high-throughput technologies has led to faster and more efficient ways of obtaining transcriptome data. RNA-seq (a sequence-based approach) has been the dominant technique in transcriptomics since the 2010s, and a recent study reported the transcriptome atlas of 16 Dezhou donkey tissues [57]. The uniqueness and specificity of chromosomal hierarchical units revealed a series of functional properties, such as cell cycle and gene regulation [58]. Hi-C technology is increasingly used in local variety formation and environmental adaptation studies [59]. It was used for haplotype-assisted assembly in the whole-genome range, and the detection efficiency and accuracy of the analysis were high [60]. Hi-C technology-based data could be widely used to identify 3D genomic structural changes due to known genomic rearrangements [61]. Hi-C technology was also used to assist with genome assembly and obtain genomes at the chromosome scale [62]. It was used for the expression regulation of genes and the study of gene function [63].

Previous studies have suggested that the Nubian wild ass and the Somali wild ass may be the ancestors of domestic donkeys. However, these findings only relied on the analysis of mtDNA variability, which only represents matrilineal inheritance. The genetic basis of the non-Dun phenotype, a pigmentation pattern that may have been selected during domestication or early post-domestication, was investigated by elucidating the history of donkey domestication through a population genomic approach that was based on whole-genome sequence data [64]. The current knowledge of donkey evolutionary history remains incomplete in the absence of archaeological and genome-wide diversity data. Thus, researchers used Hi-C technology to assemble a chromosome-level reference genome of a male Dezhou donkey from scratch and re-sequenced the genomes of 126 domestic donkeys and seven wild asses. The reduced level of Y chromosome variation was found to be inconsistent with the autosomal data, and the paternal and maternal genetic histories differed, possibly because of reproductive management. In addition, the typical staining dilution of the brown phenotype showed very similar microscopic and macroscopic features in horses and donkeys. More importantly, the same *TBX3* gene was responsible for the pigmentation of the brown phenotype in both species, whereas the mutation in the non-brown phenotype in donkeys was caused by a 1 bp deletion, which may have a regulatory role.

Hi-C technology has more advantages than the de-novo assembly from short-read data. First, Hi-C works through the spatial distance on the chromosome and the linear distance of the different resulting interaction frequencies to complete the chromosome localization. Thus, building a population is not needed because a single individual could be able to achieve chromosome localization. Second, greater marker density and more complete sequence localization could be achieved through Hi-C technology. It could generally be 90% of the above genomic sequences localized to chromosomes. Third, correcting the errors in the assembled genomic sequences is possible through the magnitude of the interaction frequency between scaffolds. 

### 5.5. Genome Assembly—Refinement Process (Donkey)

Donkey genome assembly was first performed by Huang et al. at Inner Mongolia University, China, in 2015 [53]. Sequence reads from double- and single-ended libraries were first assembled into contigs and scaffolds by using newblerv2.8. Then, longer scaffolds were constructed using SSPACE software and 37 paired library information. Finally, Gapcloser scaffolds were used to fill the gaps inside. After reassembly, the donkey genome sequence was obtained, and it consisted of 2166 scaffolds (>1 kbp), with a total size of 2.36 Gb and N50 sizes of 66.7 kb and 3.8 Mb for scaffolds. Whole-genome sequence analysis is a delicate and powerful method to study chromosome evolution. The donkey and Asian wild-ass genomes complement the reference genomes of the genus *Equus*. Comparative analyses of sequences based on these genomes could provide important insights into the population history and adaptive evolution of horses. Furthermore, these results may strengthen the understanding of chromosomal rearrangement mechanisms (using Oxford Nanopore and Pacific Biosciences technologies) and these data may contribute to studies of equine genomics and mammalian chromosome evolution. 

In 2018, Renaud et al. used emerging technologies to perform genome assembly in donkeys [55]. The difficulty with genome assembly has always been the translation of relatively short reads into longer scaffolds, with new sequencing technologies generating longer reads, often accompanied by error rates of up to 15% (e.g., Oxford Nanopore and Pacific Biosciences) [65]. Thus, single-molecule sequencing is often integrated with short reads generated by the Illumina platform to generate so-called hybrid de-novo assemblies to correct these error rates while maintaining cost effectiveness. The alternative approach, which uses long-range chromatin interactions to capture widely spaced read pairs in the genome, coupled with a custom assembly pipeline (HiRise), has been shown to produce long scaffolds at the sub-chromosomal level with low error rates (e.g., Chicago Library). When the Chicago HiRise assembly technology was used to produce high-quality genomic assemblies for donkeys, the N50 was 15.4 Mb, the N50 for contigs was 140.3 kb, and the scaffolds were four times larger than before. In addition, sex chromosomes were typically more repetitive than autosomes, and the donkey X chromosomes appeared to have undergone several rearrangements, thus limiting the N50 of the X chromosome scaffold assembled here to 0.57 Mb. This new donkey combination could be utilized by identifying the ROHs resulting from the correlation between low effective population size and ancestry, exploring chromosome rearrangements and their effect on the presence of distance patterns between donkeys and horses.

In 2020, Wang et al. constructed a de-novo assembly of the donkey genome by using cutting-edge methods [64]. This approach used Illumina short reads and PacBio long reads to create a hybrid de-novo assembly. The resulting Illumina contigs were combined with PacBio reads, and valid reads were extracted based on the results of HiC-Pro. 3d-DNA software was used to assemble chromosome-length genomes and combined with the draft PacBio assembly genome. Combination of the Hi-C data with information from the analysis of covariance among the donkey, horse, and human genomes inferred that the donkey genome is distributed in 30 autosomes, two sex chromosomes (X and Y), and one mitochondrial ring chromosome. Several evaluation methods have demonstrated high-assembly quality, showing 24-fold [53] and sixfold [55] improvements in scaffold N50 compared with previously reported donkey genomes. This assembly could facilitate the identification of small chromosomal rearrangements between horses and donkeys to clarify the evolutionary history of the equine species.

## 6. Conclusions

With the advancement of science and technology, the discovery of archaeological evidence, the development of molecular genetics, and the perfection of genome sequencing, people have increased systematic and in-depth understanding of the origin and domestication of the donkey. In this paper, the development of research on the origin and evolution of the donkey was analyzed in detail in terms of archaeology, molecular cell studies, genetic material studies, genome sequencing, and assembly, and the advanced techniques were summarized. The domestic donkey originated in Africa and has two main clades, with a single domestication in Africa about 7000 years ago, followed by further expansion in that continent and Eurasia and eventual return to Africa. Undoubtedly, assembling an individual organism by its genome and comparing it with related samples could lead to more accurate results through big data statistics, analysis, and calculation of relevant models. In the era of bioinformatics, many biological experiments could be performed by genome sequencing under the existing conditions, but large-scale and high-quality research in the field of genome assembly is still urgently needed to achieve perfect whole-genome translation, application, and editing.

## Data Availability

Not applicable.

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
