# Peer review of "Origin, Evolution, and Research Development of Donkeys"

_genes, 2022, doi:10.3390/genes13111945_

Round 1
Reviewer 1 Report
Thank you for giving me the opportunity to review this paper. As a result of the review, this paper analyzed the research and development of donkey's origin and evolution in detail in terms of archaeology, molecular cell research, genetic material research, genome sequencing, assembly, etc., and it is considered a good paper to organize the advanced techniques to get useful information about donkeys. However, I think it would be a better paper to review the latest research manuscripts such as donkey morphology, functional characteristics of meat and fat, donkey breeding characteristics, and SNP.
Author Response
Our team has also been working continuously on morphology, functional characteristics of meat and fat, donkey reproductive characteristics and SNPs, and we have published related articles. It is only that the team has mainly focused on the whole gene series and genetic breeding of donkeys in recent years, and other scientific achievements are not abundant enough to write review articles. Thus, only the origin and domestication of donkeys were reviewed from the aspects of basic information, cultural origin, bioarcheology, mitochondrial and chromosomal microsatellite sequences, and whole-genome sequence comparison in this paper. We believe that in the near future, we will review other aspects of donkeys in more detail.
Reviewer 2 Report
The article discusses the origin and evolution of donkeys from various points of view, cultural origin, archeology, molecular genetics and chromosomal analysis. At the end it makes a successful statistical analysis.
The article is relevant because it bridges a gap in the origin and evolution of donkeys, considering various methodologies of analysis, finalizing them with appropriate statistics.
This review gives important information to the industry and researchers, which could promote theoretical research, genomic selection, and reproductive management of the donkey population.
I do not recommend any improvement in relation to the methodology, but I do recommend modification of the figure - The only figure in the paper has no legend.
Author Response
Thanks for your suggestions. We have already added legend in the figure.
Reviewer 3 Report
The reviewer’s comments
This very interesting review paper describes the origin and evolution of donkeys based on a wide set of publications. In general, the paper is well written and structured, and provides valuable information for a reader interested in donkeys as well as other equine species.
My detailed comments are listed below:
· Lines 268, 270, 276, 302, 306, 357 and others: dates are given either as “years ago” or “BC”, I suggest to write all dates in a same way
· L. 337: the author's name (Stanisic et al.) is given in the first sentence of the paragraph, but the publication number is given at the end of the paragraph. This concerns also publication numbers in lines 363, 405 and others. For clarity of the text, publication numbers should be given at the beginning of each paragraph, instead of the end of the paragraph. This is especially true of the paragraph starting from 564: "In 2020, Wang et al ...", in which the publications by Huang et al., 2015 and Renaud et al., 2018 are further cited (without numbers of these articles), and finally at the end of the paragraph is given the publication number of Wang et al. Similarly, in a paragraph starting from line 545, which begins with: "In 2018, Renaud et al", without specifying the publication number; in the second sentence is given number “65”, i.e. of article by Chakraborty et al., and only at the end of the paragraph is the reference number of Renaud et al.
· L. 392: Does the sentence: "Fourteen published Y-chromosome-specific microsatellite (Y-STR) studies were performed on 395 male donkey samples from China, Egypt, Spain, and Peru /.../” refer to one publication [52] mentioned at the end of the paragraph or to 14 different publications? Please make this sentence more clearly.
· I have doubts about several chapters. It is not clear to me why point 2.3 „Microsatellite markers” is included in Chapter 2 „Research findings at the molecular cellular level”, while point 3.1. "Imprinting of fast karyotype evolution" is in Chapter 3 „Research on genetic materials”. A similar doubts arise in Chapter 4.2, "Whole genome sequencing (WGS)", in which the studies of 479 bp mtDNA control region were described in the second paragraph, although Chapter 3 seems to be more appropriate for these results.
· In my opinion, Chapter “Conclusions” should clearly summarize whether and to what extent the results obtained by different methods lead to the same conclusions about the history and origin of the domestic donkey.
· Figure at the end of the article: I do not understand the idea of adding this figure to the manuscript, without any caption, and in Italian instead of English.
· References: standardize the names of journals, giving either full names or their abbreviations. Names of journals with incorrect lower case letters should also be corrected; this concerns for example Animal genetics in item 52
Author Response
We have already uploaded a word document to response all the reviewer's comments.
